# The Role of Chitosan-Based Materials in Interactions with Antibiotics: An Overview of In Vitro and In Silico Studies

**DOI:** 10.3390/ijms262211070

**Published:** 2025-11-15

**Authors:** Joana F. M. Sousa, Dina Murtinho, Artur J. M. Valente, Jorge M. C. Marques

**Affiliations:** Coimbra Chemistry Center—Institute of Molecular Sciences (CQC-IMS), Department of Chemistry, University of Coimbra, 3004-535 Coimbra, Portugal; jfm_sousa@hotmail.com (J.F.M.S.); dmurtinho@ci.uc.pt (D.M.); qtmarque@ci.uc.pt (J.M.C.M.)

**Keywords:** antibiotics, chitosan, antibiotic-resistance bacteria, adsorption

## Abstract

The increasing presence of antibiotics in aquatic environments is a growing concern, causing ecological and public health risks. Even low concentrations of antibiotics may lead to the development of antibiotic-resistant bacteria. The interest in building new materials that can be used as templates for removing pollutants from the environment has been growing year upon year. We review the research involving adsorption processes that occur in chitosan-based materials that are employed to remove antibiotics from water. Since covering all the antibiotics that can be found in the environment would be an overwhelming task, we concentrated our efforts on describing the studies related to the removal of tetracycline, ciprofloxacin, cephalexin, and azithromycin, which are perhaps the most ubiquitous ones. We present the chemical modifications introduced into chitosan and chitosan-based materials commonly used as antibiotic adsorbents, as well as the influence of physical chemistry conditions on these processes. In addition, we also review in silico studies that have been carried out to obtain molecular-level insights into the interactions between chitosan-based adsorbents and the four mentioned antibiotics. Particular emphasis is placed on our recent computational work regarding the adsorption of tetracycline by various chitosan-based materials.

## 1. Introduction

Water is essential for life, and the provision of an adequate water supply is crucial for human well-being. Moreover, 10% of the biodiversity on our planet is found in the freshwater ecosystem. On the other hand, the pollution of water bodies has led to a deterioration of water quality and the health of all organisms living underwater [1]. Water pollution is mainly caused by the release of various substances such as household, industrial, and agricultural waste. These substances also include emerging pollutants (EPs) [2,3]. EPs can be found in soil, water bodies, food, municipal wastewater, and drinking water [4]. At least 700 substances, classified into 20 classes, have been identified in the European aquatic environment based on the NORMAN network (www.norman-network.net (accessed on 2 October 2025)).

Pharmaceuticals have become a problem due to their negative impact on ecosystems and human health; moreover, they are not monitored or regulated to protect the environment from their emissions [5,6]. These compounds are active ingredients that are classified into 24 different classes [7]. Non-steroidal anti-inflammatory drugs (NSAIDs), anticonvulsants, antibiotics, and lipid regulators are the four main classes of pharmaceuticals found predominantly in water [8].

An emerging family of environmental pollutants are antibiotics, which, despite being medicines capable of fighting bacterial infections, are used in an excessive and inappropriate way at the human, veterinary, and agricultural levels [9].

The presence of antibiotics in wastewater, such as cephalexin, has been detected in Europe in countries such as Portugal, Spain, Germany, Cyprus, Ireland, Finland, and Norway, with a maximum concentration in Finland of 308 ng/L [10]. In Portugal, Spain, Cyprus, and Germany, cephalexin was classified as a moderate environmental risk in water [10]. In China, the United States, and Australia, cephalexin was detected in water at concentrations of 980 ng/L [11], 2330 ng/L [12], and 3.900 ng/L [13], respectively. In turn, tetracycline was detected in wastewater treatment effluents in countries such as Portugal, Spain, Germany, Cyprus, Ireland, Finland, and Norway, with the highest concentration in Portugal of 613.6 ng/L [10]. It has been detected in the influents and effluents of wastewater treatment in China [14]. Among the seven European countries analyzed by Giustina et al. [10], Portugal had the highest antibiotic concentrations of ciprofloxacin and azithromycin in effluent wastewater treatment, at 1435.5 ng/L and 1577.3 ng/L [10], respectively.

In recent years, due to the pandemic caused by the SARS-CoV-2 virus, the use of antibiotics in patients with COVID-19 has increased. However, the overall use of antibiotics in humans has decreased by more than 15% in most European countries. This is mainly attributed to the low incidence of respiratory infections not associated with COVID-19, as a consequence of acquired habits to prevent the spread of COVID-19 [15,16]. The same trend was observed in the United States, except for the antibiotic azithromycin, since the use of this antibiotic was higher than expected early in the COVID-19 pandemic [17]. On the other hand, most of the water pollution caused by antibiotics is a consequence of agricultural activities. For instance, China is the world’s main consumer of antibiotics, being responsible for 52% of all antibiotics used to treat animals alone [18]. Pharmaceutical and cosmetic industry plants are another major source of pollution caused by antibiotics [19].

About 30–90% of antibiotics are excreted unchanged in urine and feces. The poor absorption of antibiotics by the body, their high and improper use, and the inadequate wastewater treatment in the pharmaceutical industry result in antibiotics ending up in the environment or municipal sewage systems [20]. The concentration of antibiotics in wastewater is too low to eliminate bacteria, but sufficient to affect the genetic and phenotypic variability of microorganisms, which may cause a rise in serious health problems due to the well-known multi-resistant bacteria [21,22,23,24].

Several steps are required for wastewater treatment: primary, secondary, and the more stringent, tertiary treatment. Primary treatment may include some treatment processes such as screening, coagulation, flocculation, and sedimentation. Secondary treatment includes conventional activated sludge, trickling filters, and clarification. Finally, tertiary treatment can include methods such as membrane filtration, advanced oxidation, reverse osmosis, membrane bioreactors, and/or adsorption [25]. The facts that tertiary processes are expensive and that current technology is not effective in removing the required amount of antibiotics from wastewater have led to great efforts to improve wastewater treatment. Among these methods, adsorption has proven to be the most viable, economical, and highly effective for antibiotic removal [26,27]. Adsorption is process that results in the transfer of individual molecules, atoms, or ions to the surface of a solid. This process is usually reversible, and the reverse process is called desorption. The adsorbed molecules are called adsorbate and the solids used to adsorb or dissolve them are called adsorbents. They can interact with each other by physical or chemical forces [28]. The adsorption efficiency directly depends on the physiochemical properties of the adsorbents and adsorbates and other environmental factors, such as the presence of other pollutants/interferents, for example, metal ions [29]. Currently, commercialized adsorbents are employed to remove antibiotics from wastewater by adsorption [30], which include carbon-based materials (activated carbon [31], carbon nanotube [32], and graphene [32]), zeolites [33,34], ion exchange resins [35], and clays [36]. However, they have high costs and potential sustainability issues. Recently, polysaccharides have been used as adsorbents due to their high adsorption capacity, low cost, renewability, biodegradability, ease of modification, and biocompatibility [37]. They have been used to adsorb, for example, heavy metals [38], dyes [39], phenols [40], oils [41], and pharmaceuticals [37]. Polysaccharides represent a class of carbohydrate polymers linked by glycosidic bonds that, according to the originated sources, can be divided into animal polysaccharides (chitin, chitosan, hyaluronic acid, and chondroitin sulphate), plant polysaccharides (cellulose, pectin, alginate, agarose, carrageenan, guar gum, konjac glucomannan, starch, and cyclodextrins), microbial polysaccharides (pullulan, dextran, salecan, gellan gum, and xanthan gum), and algae polysaccharides (alginic acid and κ-carrageenan) [42]. This review focuses on the application of chitosan and its derivatives for the adsorptive removal of antibiotics such as tetracycline, cephalexin, ciprofloxacin, and azithromycin from wastewater. In this article, experimental and computational studies will be discussed. The paper is organized as follows: Section 2 summarizes the main classes of antibiotics. Section 3 and Section 3.1 discuss the chemical modification process of chitosan-based adsorbents and examples of chitosan-based materials, respectively. The influence of physical chemistry parameters on antibiotic adsorption (Section 4.1, tetracycline and Section 4.2, ciprofloxacin) is presented in Section 4. In Section 5, computational studies are discussed, while Section 6 presents the main conclusions of this review along with a future outlook.

## 2. Antibiotics

Antibiotics are chemical substances that inhibit or eliminate the growth of microorganisms such as bacteria, protozoa, or fungi and can be divided into several classes (e.g., β-lactams, sulfonamides, quinolones, tetracyclines, macrolides, and others) [43]. Global antibiotic consumption has increased by 46% over the last two decades. North Africa, the Middle East, and South Asia have seen the largest increases, at 111% and 116%, respectively [44]. In Europe, the main antibiotics consumed are penicillins, macrolides, tetracyclines, sulfonamides (United Kingdom and Italy) [45], and cephalosporins (https://www.ecdc.europa.eu/en/antimicrobial-consumption (accessed on 2 October 2025)) [46]. Among second-generation cephalosporins, cefuroxime accounts for 50% of antibiotic consumption. Cefalexin is the most commonly used first-generation cephalosporin, while cefixime, cefpodoxime, and ceftriaxone are the most commonly used third-generation cephalosporins in Europe [47].

A summary description of the main classes of antibiotics and their medical application is presented in Table 1.

From the analysis of the literature, we can see that a significant number of published studies related to antibiotic adsorption mainly involve tetracycline and, to a lesser extent, ciprofloxacin, cephalexin, and azithromycin.

Tetracyclines are formed by an inflexible skeleton of four benzene rings, with many functional groups such as alkyl, hydroxyl, and amine on the upper and lower sides of the molecules (Figure 1A(a–d)). The presence of the amino group in the C4 position is crucial for antibiotic activities [48]. Tetracycline is characterized by three different pK_a_ values (3.3, 7.7, and 9.7) and exhibits optimal antimicrobial activity in the pH range of 5.5–6.0 [49], which can be justified by its zwitterionic structure. Although they have known adverse effects, including gastrointestinal disturbances and photosensitivity, tetracyclines continue to be widely prescribed due to their effectiveness against a broad spectrum of bacteria.

Quinolones are a class of heterocyclic compounds and constitute one of the main classes of antibiotics. They are structurally characterized by exhibiting a quinoline nucleus with a carboxyl group at position 3 and a carbonyl group at position 4 of the bicyclic core [50]. Ciprofloxacin (Figure 1B(a–c)), in particular, is an amphoteric compound with different pK_a_ values of 6.09 and 8.74 [51] and a high water solubility of 36 mg/mL (logP = 0.28) [52]. Ciprofloxacin was the first compound in this family to show effective systemic activity against most strains of Gram-negative bacteria and certain Gram-positive bacteria. In addition, ciprofloxacin is used in the treatment of other diseases, including low-risk febrile neutropenia, and as a second-line treatment for cholera [53]. This action of ciprofloxacin is related to the cyclopropyl group present in its structure [54].

Cephalexin (Figure 1C(a–c)) is a first-generation cephalosporin antibiotic, named as 7-(D-α-amino-α-phenylacetamido)-3-methyl-3-cephem-4-carboxylic acid monohydrate. Cephalexin has a logP = 0.65 and shows two pK_a_s at low pH—in the range of 2.34 and 3.11 due to the carboxylic group, and another one in the range of 6.62 and 7.18 as consequence of the occurrence of an ammonium group [55,56]. Cephalexin can be used against Gram-positive and Gram-negative microorganisms; however, high efficacy occurs only against different strains and isolates of *Streptococci* and *Staphylococci*, which makes this β-lactam antibiotic less widely used than the previous ones [57].

Azithromycin is a broad-spectrum antibiotic from the third-most-consumed group of antibiotics, the class of macrolides (Table 1). Azithromycin contains a 15-membered lactone ring with a nitrogen atom in the macrocycle (called azalide) and a methyl-substituent in this nitrogen which allows for the prevention of its metabolism [58]. The broad-spectrum effect is essentially due to the effective accumulation in phagocytes and, consequently, the finding of an effective vector to reach the infection sites. Additionally, it also acts by reducing the formation of biofilms and mucus production. The latter allows for the extension of the time for antibacterial action [59].

**Table 1 ijms-26-11070-t001:** Classes of antibiotics and medical uses.

Classes	Antibiotics	Medical Uses
β-lactam	Penicillins amoxicillin, ampicillin, penicillin G, oxacillin, cloxacillin Cephalosporins First-generation: cefazolin, cephalothin, cephapirin, cephradine, cefadroxil, and cephalexin. Second-generation: cefuroxime, cefmetazole, cefotetan, and cefoxitin. Third-generation: cefotaxime, ceftazidime, cefdinir, ceftriaxone, cefpodoxime, and cefixime. Fourth-generation: cefepime, cefozopran, cefpirome, and ceftaroline.	Used in the prevention and treatment of bacterial infections, such as otitis media, prophylaxis, respiratory infections, and skin and urinary infections, caused by Gram-positive and Gram-negative bacteria [60].
Sulfonamides	Sulfamethoxazole, sulfisoxazole, sulfacetamide, sulfadiazine, sulfadimethoxine, sulfaguanidine, sulfamerazine, sulfamethazine, sulfapyridine, and sulfathiazole.	Wide-spectrum drugs active against a range of bacterial species, both Gram-positive and Gram-negative. They work by interfering with the synthesis of folic acid in bacteria. They can be used to treat gastrointestinal and respiratory tract infections [61].
Macrolides	Clarithromycin, erythromycin, roxithromycin, azithromycin, tylosin, and clindamycin	Used successfully in the treatment of infectious diseases in humans and animals involving the upper and lower respiratory tract, the skin, and skin structures. These antibiotics are active against aerobic and anaerobic Gram-positive bacteria [62,63].
Tetracyclines	First-generation: tetracycline, oxytetracycline, and chlortetracycline. Second-generation: doxycycline and minocycline. Third-generation: tigecycline, omadacycline, and eravacycline	Activity against a wide range of microorganisms including Gram-positive and Gram-negative bacteria, chlamydiota, mycoplasmatota, rickettsiae, and protozoan parasites. Used to treat bacterial infections of skin, intestines, respiratory and urinary tracts, lymph nodes, etc. [48]
Quinolones	First-generation: Nalidixic acid and pipemidic acid. Second-generation: Ciprofloxacin, Ofloxacin, Norfoxacin, Lomefloxacin, Enoxacin, and Fleroxacin. Third-generation: levofloxacin, gatifloxacin, and sparfloxacin Fourth-generation: trovafloxacin, moxifloxacin, and gemifloxacin	Often used for genitourinary infections and are widely used in the treatment of hospital-acquired infections associated with urinary catheters and pneumonia [64].

## 3. Chitosan-Based Adsorbents

### 3.1. Chemical Modification of Chitosan for Adsorption Applications

The success of adsorption as a technique for removing pollutants, such as antibiotics, from wastewater is related to, among other factors, the numerous materials available. Although polysaccharides are considered good adsorbents when compared to other types of adsorbents, they have some disadvantages, such as difficult regeneration and reutilization and weak mechanical properties. For this reason, chemically modified polysaccharides have been developed to create new biomaterials with unique physicochemical properties for the removal of antibiotics from water and wastewater.

Among polysaccharides, chitosan—a cationic polysaccharide obtained from the deacetylation of chitin—is one of the most-used for different applications. The structural unit of chitin is *N*-acetylglucosamine, an amide derivative of glucose, which is usually extracted from seafood processing wastes. It is the second-most-abundant biopolymer in nature, after cellulose [37]. Due to its many free hydroxyl and amino groups (Figure 1D(a,b)), electrostatic attraction and hydrogen-bonding leads chitosan to be a good adsorbent. Chitosan also has the advantage of being very abundant, non-toxic, biodegradable, readily renewable, low-cost, easily modified, and reusable. These characteristics make chitosan the most widely used animal polysaccharide for adsorbents [37]. However, it has poor mechanical properties, such as rigidity and brittleness, it is poorly soluble in many solvents, and unstable in acidic medium. It is important to note that the degree of deacetylation and the polymer’s molecular weight are significant factors impacting the final physical properties of chitosan. Chitin deacetylation can be achieved through enzymatic processes or hydrolysis. The most common method for this purpose is hydrolysis in a basic medium, and the degree of deacetylation can be modeled by controlling the reaction conditions: base concentration, temperature, and reaction time. Higher degrees of deacetylation increase the content of primary amine groups, consequently altering the polymer’s physical and mechanical properties. The origin of the chitin and the treatments it undergoes can lead to the production of chitosan with different molecular weights, which also have a significant impact on the final properties of the material. For instance, very-high-molecular-weight chitosan is difficult to dissolve, resulting in highly viscous solutions, which makes the processing of the polymer difficult [65].

To solve these problems, chemical modification processes such as cross-linking and grafting have been proposed for chitosan [66].

The chemical modifications that are usually performed on chitosan involve reactions of the amine groups and/or hydroxyl groups of the glucosamine ring. When the aim is to obtain materials that can be used as adsorbents, cross-linked polymers are commonly prepared to improve the polymer’s mechanical properties and the adsorbates’ absorption capacity (through the formation of highly porous structures or structures with different degrees of hydrophobicity/hydrophilicity). Cross-linking can be induced by the application of physical stimuli (such as temperature, radiation, and pH), by the formation of hydrogen bonds or ionic bonds, among others (physical cross-linking). Cross-linking agents such as glycerophosphates or sodium tripolyphosphate have been widely used to promote the ionic cross-linking of chitosan.

Whenever covalent bonds are formed, the cross-linking is referred to as chemical. Compounds with aldehyde groups have been extensively used as chemical cross-linkers for chitosan because they react easily with amine groups to form imines. If formaldehyde is used to promote the reaction, the imine formed may also react with another amine group from another chitosan molecule, forming amines. Dialdehydes such as glutaraldehyde, glyoxal, or terephthalaldehyde form imines by reaction with chitosan, so they are also broadly used as cross-linking agents for this polymer [67].

Epoxides such as epichlorohydrin or ethylene glycol diglycidyl ether constitute another class of chemical cross-linkers for chitosan. In this case, cross-linking occurs through the reaction of the polymer’s hydroxyl or amine groups with the epoxide groups, resulting in ring opening. In the case of epichlorohydrin, a chlorine substitution reaction occurs in addition to ring opening, which enables chitosan cross-linking.

Di- or polyacids can react with the free amine groups of chitosan to form amides (chemical cross-linking), or they can form ionic bonds with these groups to promote physical cross-linking. Other compounds such as diisocyanates and *N*,*N*′-methylenebisacrylamide are also used as cross-linking agents for chitosan.

However, some of these compounds are associated with toxicity issues. Therefore, whenever possible, using cross-linking agents that are less harmful to living organisms and the environment, such as genipin or natural acids (malic, citric, succinic acids, etc.), is a more sustainable option [68,69].

Cross-linked chitosan-based polymers have been used as adsorbents for the removal of hazardous compounds, including pharmaceuticals [65].

The preparation of graft copolymers is another strategy that can be used to improve the properties of chitosan, including solubility, stability, biological activity, and compatibility with synthetic polymers, among others. Chitosan graft copolymerization is typically carried out with redox radical initiators, such as potassium or ammonium persulfate, cerium ammonium nitrate, and ferrous ammonium sulfate. However, enzymes and irradiation, including gamma and microwave radiation, can also be used to promote the initiation. A wide range of monomers can be used to prepare graft copolymers, as described in the literature. The choice of the monomer is made according to the intended application. The most-used monomers include acrylamide and its derivatives, methyl methacrylate, acrylic acid, methacrylic acid, and acrylonitrile. The extent of the copolymerization reaction, namely the number and size of the grafted chains, and consequently the properties of the materials obtained, is determined by parameters such as initiator and monomer concentrations, temperature, and reaction time. Chitosan graft copolymers have been used in drug delivery systems, as antimicrobial agents, and also in wastewater treatment for the adsorption of pollutants [70].

Although cross-linking or grafting improves the physical and mechanical properties of chitosan, it is sometimes necessary to combine it with other polysaccharides or other materials to create blends or composites that meet the requirements of a given application.

Below, we will describe and analyze the most relevant examples of the synthesis and application of chitosan-based matrices as adsorbents, as seen in Figure 2.

### 3.2. Chitosan-Based Materials

Studies reported in the literature on chitosan-based materials used for the adsorption of antibiotics, specifically tetracycline, ciprofloxacin, cephalexin, and azithromycin, were compiled. The following subsections categorize these materials according to their structural characteristics and their capacity for adsorption. The main findings are summarized in Table 2 and Table 3, corresponding to tetracycline and ciprofloxacin, respectively.

**Table 2 ijms-26-11070-t002:** Adsorption of tetracycline onto chitosan composites: interactions, maximum amount of adsorption (q_max_), and pH values from water.

Adsorbent	Adsorption Interactions	*q*_max_ (mg/g)	pH	Ref.
Nitrilotriacetic acid-modified magnetic chitosan microspheres (NDMCMs)	π-π interactions, cation-π bond, hydrogen bond, and amidation reaction	373.5	8	[71]
Fe_3_O_4_@SiO_2_-Chitosan/GO (MSCG)	Electrostatic interactions and π-π interactions. Cu(II) also acts as a bridge	n.r.	6–7	[72]
Chitosan, thiobarbituric acid, malondialdehyde, and Fe_3_O_4_ nanoparticles (CTM@Fe_3_O_4_)	π-π and hydrogen-bonding interactions	215.31	7	[73]
CuCoFe_2_O_4_@Chitosan	Electrostatic interaction	n.r.	3	[66]
Carbon disulfide-modified magnetic ion-imprinted chitosan-Fe(III), (MMIC-Fe(III) composite) ^1^	Electrostatic interactions and hydrogen bond interactions. Tetracycline or Cd(II) could act as a bridge	516.29	8	[74]
Functionalized zero-valent iron/walnut shell composite (CS-WS-NZVI)	n.r.	n.r.	6	[75]
Chitosan, diphenylurea, and formaldehyde with magnetic nanoparticles (MnFe_2_O_4_), (CDF@MF)	van der Waals forces, π–π stacking interactions, and hydrogen-bonding interactions	168.24	6	[76]
Biochar modified by Chitosan and Fe/S (BCFe/S)	Electrostatic attraction, π-π stacking, pore filling, silicate bonding, hydrogen-bonding, chelating, and ion exchange	183.01	5	[77]
Calcined chitosan (CS)-supported layered double hydroxides ^2^	Electrostatic interactions	195.31	9	[78]
Magnetic chitosan-*g*-poly(2-acrylamide-2-methylpropanesulfonic acid) (CTS-*g*-AMPS) porous adsorbent ^3^	Electrostatic attraction, hydrogen-bonding	806.60	3	[79]
Glutaraldehyde-cross-linked electrospun nanofibers of chitosan/poly(vinyl alcohol) (GCCPN)	Electrostatic interactions	102	6	[80]
Na-montmorillonite (Na-Mt) with carboxymethyl-chitosan (CMC) (CMC-Mt) ^3^	Electrostatic interactions	48.10	4–6	[81]
NiFe_2_O_4_-COF-chitosan-terephthalaldehyde nanocomposites film (NCCT) ^4^	Cation exchange, electrostatic attraction, hydrogen-bonding, and π–π interactions	246.41	8	[82]
Chitosan–kaolin (Cs-k-Fe_3_O_4_)	n.r.	28.3	5–8	[83]
Zinc ferrite/chitosan–curdlan (ZNF/CHT-CRD)	Electrostatic interactions, hydrogen bond, Yoshida, and dipole–dipole hydrogen and cation-π interactions	371.42	6	[84]
Magnetic chitosan (CS·Fe_3_O)	Hydrogen bonds and cation–π interactions	211.21	7	[85]
MIL101(Fe)/ZnO chitosan	π -π interactions	31.12	6.55	[86]
ZIF-8-chitosan composite	Electrostatic interaction, π-π stacking interaction, and hydrogen-bonding	495.04	9	[87]
Chitosan-supported layered double hydroxide calcined at 400 °C (CSLDO400) ^2^	Electrostatic interactions	195.31	9	[88]
Magnetic nanocomposite (CS/N-zeolite/Fe_3_O_4_/MOF-808)	Electrostatic interaction, π-π stacking interactions and hydrogen-bonding, cationic-π interactions, and Lewis acid–base interactions	31.69	7	[89]
Chitosan-functionalized MIL-101@CNM (Ch/MIL-101@CNM) composites	Electrostatic interaction, π-π stacking interactions, and hydrogen-bonding	33.77	6.51	[90]
Biochar derived from chitosan modified with ammonium persulfate (APS) and acetic acid (APS@CHI-1:3) ^3,5^	Electrostatic interactions, π-π stacking interactions, hydrogen-bonding, and Lewis acid–base interactions	851.5	5	[91]

n.r.: not reported. Adsorbate: tetracycline and ^1^—Cd(II), ^2^—methyl orange, ^3^—chlortetracycline, ^4^—cefotaxime, and ^5^—Cr(VI).

**Table 3 ijms-26-11070-t003:** Adsorption of ciprofloxacin onto chitosan composites: interactions, maximum amount of adsorption q_max_, and pH values from water.

Adsorbent	Adsorption Interactions	*q*_max_ (mg/g)	pH	Ref.
Chitosan/biochar hydrogel beads (CBHB)	π-π electron donor–acceptor (EDA) interactions, hydrogen-bonding, and hydrophobic interactions	76.00	3	[92]
Chitosan–biochar beads (CH-BB) ^1^	n.r.	1.49 ± 0.06	n.r.	[93]
Titanium-biochar/chitosan hydrogel beads (TBCB)	n.r.	50.916	9	[94]
Humic acid-biochar/chitosan hydrogel beads (HBCB)	Hydrogen-bonding, π-π electron donor–acceptor (EDA) interactions, and hydrophobic interactions	154.89	8	[95]
Chitosan-grafted SiO_2_/Fe_3_O_4_ (Chi-SiO_2_/Fe_3_O_4_)	n.r.	100.74	12	[96]
Sodium lignosulfonate/chitosan@ZIF-8 (SLS/CS@ZIF-8)	Electrostatic interactions, hydrogen-bonding interactions, and π-π interactions	413	6–9	[97]
Fe_3_O_4_@MIL101(Fe) chitosan composite beads ^2^	π-π interactions, H-bonding, and electrostatic interaction	31.30	9	[98]
Magnetite-imprinted chitosan polymer nanocomposites (Fe-CS NCs)	Electrostatic interactions and hydrophobic interactions	142	6.5	[99]
Nanocomposites of hydroxyapatite (HAP), chitosan (CT), and magnetite (MNP)	n.r.	0.517	n.r.	[100]
Aluminosilicates-grafted chitosan (Al_2_O_3_@SiO_2_-chitosan)	n.r.	31	5.77	[101]
Magnesium oxide, chitosan, and graphene oxide (MgO/Chit/GO) nanosheets ^3^	Electrostatic interactions and π-π interactions	1111	7	[102]
Zn(II)-impregnated chitosan/graphene oxide composite (Zn(II)-CS/GO)	Electrostatic interactions and π-π interactions	210.96	6.5	[103]
Superparamagnetic (Fe_3_O_4_-MoS_2_@CS) nanomaterials ^5^	Electrostatic interactions and hydrophobic interactions	190.7	7	[104]
Granular hydrogel prepared from chitosan as the grafted backbone and acrylic acid as the polymerizable monomer (3D structured hydrogel CTS-PAA) ^4^	Electrostatic interactions	267.7	3	[105]
Chitosan-derived carbon–smectite nanocomposite with cobalt (H_Co/C-S)	Electrostatic interactions, hydrogen-bonding, and π-π interactions	72.3	6	[106]
EDTA-functionalized β-cyclodextrin-chitosan (β-CD-CS-EDTA) composite) ^6^	Chelation, electrostatic interactions, and host–guest inclusion interactions	25.40	4.30	[107]
Magnetic chitosan@Ag-MWCN nanocomposite	Electrostatic interactions, hydrogen-bonding, and π-π interactions	31.26	9	[108]
5/Ag/BZO/Cht composites	n.r.	95	9	[109]
Magnetic selenium-based metal–organic framework (MSe-MOF) incorporated into chitosan/alginate biopolymer hydrogel beads (MSCA)	Electrostatic interactions, hydrogen-bonding, ion exchange, and pore filling	440	8	[110]

n.r.: not reported. Adsorbate: ciprofloxacin and ^1^—metal(loid)s As, Cd, and Ps, ^2^—tetracycline and doxycycline, ^3^—norfloxacin, ^4^—enrofloxacin, ^5^—Cr(IV), and ^6^—Ph(II), Cu(II), and Ni(II).

#### 3.2.1. Chitosan-Based Polymer Composites

Taking advantage of the large amounts of corn stalk waste available, Lei et al. [111] have extracted various polysaccharides from the waste and used them to prepare blend hydrogels. In this process, a hydrogel containing cellulose and chitosan, cross-linked with *N*,*N*′-methylene-bis-acrylamide, was prepared and its performance for TC adsorption was evaluated. From all the synthesized hydrogels, which included sodium alginate and pectin, among others, the cellulose–chitosan hydrogel showed the best adsorption capacity for tetracycline, although with a moderated removal efficiency (ca. 20%), which was justified by its higher specific surface area and pore volume when compared to other hydrogels [111].

Electrospun nanofibers have been generated by a variety of different polymers such as poly(ethylene oxide), poly(vinyl alcohol) (PVA), poly(lactic acid), and poly(caprolactone) [112]. The advantage of nanofibers is that they have a high specific surface area, high porosity, and small pore size, essential for adsorbents to achieve a high degree of adsorption. Abdolmaleki et al. [80] have prepared electrospun nanofibers of chitosan/PVA, cross-linked with glutaraldehyde, for the adsorption of tetracycline, achieving a maximum adsorption capacity of 102 mg/g [80]. Whilst PVA tends to increase the mechanical and elastic strength of the blend, the chemical cross-linking can be used not only to improve its mechanical and chemical properties but also to increase its surface area. Different cross-linking agents as, for example, glutaraldehyde, epichlorohydrin, and polyphosphates, have been used [113]. However, as a drawback, the cross-linking leads to a decrease in the number of available chitosan amino groups and, consequently, the number of active sites for adsorption also decreases. The ability to capture pollutants depends on the number of functional groups available on the surface of the adsorbent. One way to overcome this limitation is the functionalization of chitosan by using, for example, 2-acrylamido-2-methyl-1-propanesulfonic acid and acryloyloxyethyl trimethyl ammonium chloride [114,115]. Bioinspired chitosan/polyvinyl alcohol (PVA) composites beads cross-linked with layered triple hydroxide (LTH)-doped bacterial cellulose hydrochar was developed to remove azithromycin. To improve the porous structure and surface functionalities, layered triple hydroxide (LTH)-doped bacterial cellulose hydrochar was incorporated into the adsorbent. The maximum adsorption capacity of this material was 1845 mg/g for azithromycin [116]. Overall, polymer blends and hydrogel formation can improve the physical properties and porosity of chitosan-based adsorbents. However, excessive cross-linking can reduce the number of available amino groups; therefore, it is necessary to strike a balance between structural reinforcement and active-site accessibility.

#### 3.2.2. Magnetic and Porous Composites

As with most adsorbents, the separation of chitosan from water after adsorption is not a simple process using conventional methods. Thus, a simple but effective strategy is to combine chitosan with magnetic nanoparticles. The use of these particles to produce magnetic chitosan composites overcomes this drawback and allows for the easy magnetic separation of the composites in the presence of an external magnetic field [117]. Magnetic chitosan (CS.Fe_3_O_4_) showed efficient adsorption of tetracycline, as the highest maximum adsorption capacity reported was equal to 211 mg/g [85]. Among the categories of magnetic nanoparticles, ferrite spinels (e.g., ZnFe_2_O_4_ and MnFe_2_O_4_) are also commonly used due to their high surface area, thermal and chemical resistance, and magnetic strength [66,76]. The incorporation of ZnFe_2_O_4_ into chitosan and curdlan gum led to adsorption values of around 371 mg/g for tetracycline [84]. On the other hand, MnF_e2_O_4_ has been used to prepare a porous magnetic nanocomposite as an adsorbent. The synthesis was carried out through condensation using chitosan, diphenylurea, and formaldehyde in combination with MnFe_2_O_4_. The maximum adsorption capacity was 168 mg/g towards tetracycline, and the regeneration study showed an adsorption performance of 158% after seven cycles [76]. Another strategy involves grafting chelating agents onto magnetic chitosan microspheres. For example, nitriloacetic acid, a chelating agent with three carboxyl and a tertiary amine groups, can be used and grafted onto the surface of magnetic chitosan microspheres, leading to a maximum adsorption capacity for tetracycline of 373.5 mg/g [71]. However, silica can also be used to improve the performance of magnetic particles [96]. Silica also has other advantages such as stability in acidic medium and inertness to redox reactions. For example, ciprofloxacin removal was observed using aluminosilicates-grafted chitosan (Al_2_O_3_@SiO_2_-chitosan), which was prepared by the sol–gel process [101]. Other studies prove that the addition of Ag ions into the structure of the adsorbents improves the adsorption of ciprofloxacin [108,109]. Clay-based adsorbents have also been used for the removal of antibiotics from water. A smectite–chitosan-derived nanocomposite containing cobalt was synthesized to remove ciprofloxacin [106].

The preparation of cellular polymers with a well-defined and high porosity can be achieved by using a high-internal-phase emulsion (HIPE) template method [118,119,120]. HIPE-formed matrices exhibit high porosity, tunable pore size, and sufficient functional groups in porous polymer materials. An example of using this method is the grafting of poly(2-acrylamine-2-methylpropanesulfonic acid) onto chitosan in an Fe_3_O_4_-stabilized pickering-HIPE, having resulted in an adsorption capacity of 807 mg/g for tetracycline [79].

In order to remove azithromycin using a chitosan-based adsorbent, Azari et al. developed a magnetic NH_2_-MIL-101(AI)/chitosan nanocomposite adsorbent (MIL/Cs@Fe_3_O_4_ NCs). The highest adsorption efficiency occurred at pH = 7.992, with a 98.362% removal efficiency. The process of adsorption occurs by forming a monolayer, which can be justified by the ionic interactions and hydrogen-bonding occurring between azithromycin and MIL/Cs@Fe_3_O_4_NCs [121]. A different strategy was developed by Li et al. [122] for the removal of cephalexin from water; they initially prepared magnetic composites by coating the chitosan layer containing γ-Fe_2_O_3_ nanoparticles on the surface of the yeast; Then, magnetic molecularly imprinted polymers were synthesized by atom-transfer radical polymerization and used to selectively recognize CFX molecules [122].

Other porous materials that have attracted attention in the efficient adsorption of antibiotics from wastewater are metal–organic frameworks (MOFs). They are characterized by an adjustable pore size, a large surface area, high porosity, high stability, recyclability, and ease of functionalization [123]. Some studies have reported the synthesis of MOFs and chitosan beads for the removal of tetracycline [86,87,89,90]. For example, Zhao et al. [124] incorporated zeolite imidazole frameworks-8 (ZIF-8) into a chitosan matrix using metal hydroxide/chitosan composite beads (ZIF-8-chitosan composite beads) as a MOF precursor. The maximum adsorption capacity of this material was 495 mg/g for tetracycline [124]. ZIF-8-chitosan has also been evaluated for the removal of ciprofloxacin from wastewater [97]. In the latter, in order to improve the availability of functional groups, lignin was added to the composite, which improved not only the adsorption capacity of ciprofloxacin but also its mechanical strength [97]. Hydrogel beads of magnetic selenium-based metal–organic frameworks (MSe-MOFs) incorporated into chitosan/alginate were developed for ciprofloxacin removal, achieving a maximum adsorption capacity of 440 mg/g [110]. Magnetic nanoparticles and porous materials such as clays, silicates, or MOFs have significantly increased the adsorption efficiency and reusability compared to the neat chitosan. Nevertheless, these systems still face challenges related to synthesis complexity and stability in real wastewater environments [84,123,124].

#### 3.2.3. Biochar and Graphene Oxide

Biochar is a carbon-rich material obtained through the pyrolysis of biomass and constitutes a low-cost and environmentally friendly adsorbent with a relatively large surface area and abundant surface functional groups. It has been used in many environmental applications, including the adsorption of a large variety of pollutants [92,93]. For example, for the adsorption of TC, composites of chitosan and ferric salt with different percentages of FeSO_4_ have been prepared. It was found that the composites with higher FeSO_4_ contents showed the highest adsorption capacity for tetracycline (183 mg/g) [77]. However, biochar has few functional groups, and it is difficult to separate from a suspension. To improve separation, durability, strength, and absorption efficiency, biochar was encapsulated within a chitosan polymeric network [92,93,94]. One possible strategy to overcome this limitation is to introduce additional functional groups on the surface by using humic acid. Humic acid-biochar/chitosan hydrogel beads have been tested for the removal of ciprofloxacin, and an increase in the adsorption capacity for the antibiotic was observed when compared to the composite without humic acid treatment [95]. Recently, Wu et al. [91] developed a material-based biochar with a high specific surface area (1656 m^2^/g) and abundant surface functional groups to remove tetracycline. Biochar was derived from chitosan modified with ammonium persulfate and acetic acid, through a two-step carbonization process: hydrothermal pre-carbonization at 180 °C for 6 h, followed by K_2_CO_3_ activation (1.3 mass ratio) at 800 °C for 2 h under N_2_. The adsorption capacity was 851.5 mg/g for tetracycline [91].

To improve the adsorption capacity and removal efficiency of chitosan, nanocomposites containing graphene oxide [125] were prepared. Recently, nanosheet composites prepared from magnesium oxide, chitosan and graphene oxide (MgO/Ch) [102], and Zn(II)-impregnated chitosan/graphene oxide composite (Zn(II)-CS/GO) [103] were prepared for the removal of ciprofloxacin. The (MgO/Chit/GO) composite demonstrated a high performance for the removal of ciprofloxacin with a maximum adsorption capacity of 1111 mg/g [102]. As an alternative to graphene, two-dimensional transition metal dichalcogenides, such as molybdenum disulfide, were used as absorbents to remove ciprofloxacin. Due to the abundance of amino functional groups on the chitosan surface and active sulfur groups on MoS_2_, Fe_3_O_4_-MoS_2_@CS materials demonstrated a high removal capacity for ciprofloxacin and chromium [104]. Other chitosan-based adsorbents were developed to remove ciprofloxacin and heavy metals that included β-cyclodextrin and ethylenediaminetraacetic acid (EDTA). However, these adsorbents demonstrate better adsorption for heavy metals than for ciprofloxacin [107,126]. Composites of the carbon-based chitosan demonstrated some of the highest adsorption capacities reported, particularly with regards to tetracycline and ciprofloxacin. However, they have some problems, such as synthesis costs [125].

#### 3.2.4. Regeneration and Reuse of Chitosan-Based Composites

A critical aspect related to the practical feasibility of adsorption processes is the fate of the adsorbent after antibiotic uptake. Several studies have addressed this issue by performing adsorption–desorption cycles in which the adsorbents are regenerated and reused. In general, chitosan-based adsorption shows that regeneration is possible using mild chemical eluents. Many studies have demonstrated excellent efficiency after some adsorption–desorption cycles (five cycles on average) compared to the adsorbent without chitosan, suggesting that the additional functionality provided by chitosan can enhance stability and resistance to degradation [85,90,97,105]. However, adsorption efficiency declines over successive cycles, and issues such as the irreversible binding of antibiotics molecules, blockage of active sites, or degradation of the polymeric matrix are cited as possible causes [90,92,97,115]. Nonetheless, regeneration procedures can introduce additional costs, solvent consumption, and the potential release of adsorbed antibiotics, which remain important limitations for large-scale implementation. To ensure sustainability, future investigations should therefore focus on improving regeneration efficiency and adsorbent durability, as well as on developing environmentally safe strategies for the disposal or valorization of spent materials.

## 4. Dependence of Adsorption on the Physical Chemistry Parameters

### 4.1. Tetracycline

The pH of the solution has a significant impact on the adsorption performance of antibiotics due to the acid–base features of most antibiotics and because it affects the surface charge of the adsorbent. Therefore, a good balance between these two factors will deeply affect the adsorbate–adsorbent equilibrium and, consequently, the adsorption capacity [66].

Taking as example the adsorption process of TC on ZIF-8-chitosan composite beads, it can be found that the adsorbent point of zero charge (*pzc*) is equal to 9.6. At pH > 9.6, the removal capacity significantly decreases due to the adsorbent–adsorbate electrostatic repulsion [124]. When the pH ranges between 7.7 and 9.6, the adsorption capacity increases because of the positive surface charge of the adsorbent. Two different interactions play an important role: those occurring between tetracycline and chitosan through the -NH_2_ group, and those between tetracycline and ZIF-8 through the imidazole ring. However, there are other factors that drive the adsorption of tetracycline, including hydrogen bonds that occur between the groups -OH, -NH_x_ as H-receptors in tetracycline, and the O-containing and *N*-containing groups in ZIF-8 and chitosan, as well as π-π stacking interactions between tetracycline and ZIF-8 through the imidazole ring [124]. A similar pattern holds for some other studies in which maximum adsorption was achieved at a pH of 8 [71,74,82].

However, other interactions can be responsible for the adsorption process. In the case of the NiFe_2_O_4_-COF-chitosan-terephthalaldehyde nanocomposites film (NCCT), characterized by a pH*pzc* = 8.6, the adsorption of TC is also driven by hydrogen bonds that can be formed between the hydroxyl groups of NCCT and tetracycline, as well as π-π interactions between NCCT and tetracycline, highlighted by the shifting peak of C=C from 1582 cm^−1^ to 1597 cm^−1^. The four aromatic rings of tetracycline can be bound to the free π-electrons of NCCT due to the aromatic ring skeleton of the COF shell and terephthalaldehyde. Complexation and cation exchange can also occur. The shifted -OH peaks from 3436 cm^−1^ to 3428 cm^−1^ suggest cation exchange of NCCT with deprotonated tetracycline, and the shifted Fe-O stretching vibration could indicate surface complexation of iron atoms with the species of tetracycline [82]. In the carbon disulfide-modified magnetic ion-imprinted chitosan-Fe(III), (MMIC-Fe(III)) composite, the complexation reactions between the deprotonated tricarbonylamide and the phenolic groups of tetracycline and positively charged Fe(III) lead to an increase in tetracycline adsorption [74]. In contrast to the other two studies, the pH*pzc* for the nitrilotriacetic acid-modified magnetic chitosan microspheres (NDMCMs) is acidic and equal to 3.7. The main reason for the adsorption of tetracycline was the electrostatic attraction between the anionic tetracycline and the tertiary amino groups of nitrilotriacetic acid on the surface of the adsorbent. The adsorption mechanism of tetracycline also involves π-π interaction, cation-π bonding, and hydrogen-bonding.

The maximum adsorption for tetracycline was observed in neutral solutions, in the pH range of 6–7, in a large number of studies [72,73,76,81,86]. Tetracycline is mostly present in a zwitterionic form, and its adsorption by adsorbents can be due to electrostatic interactions, hydrogen-bonding, and π-π interactions. The Fe_3_O_4_@SiO_2_-Chitosan/GO (MSCG) nanocomposite was studied with and without Cu(II), to investigate the electrostatic interactions that occur between tetracycline and the positively charged nitrogens in (MSCG). When Cu(II) was present, it participated in a bridging interaction between tetracycline and the adsorbent. Cu(II) interacted with the C=O and -OH groups of tetracycline and then formed a strong complex with the amino groups of the absorbent. π-π interactions also occur between tetracycline and the absorbent through the bulk π system on the graphene oxide surface and the benzene rings of tetracycline [72]. π-π-electron-donor–acceptor interactions also occur between the aromatic ring of chitosan, diphenylurea and formaldehyde with magnetic nanoparticles (CDF@MF), and the benzene rings of tetracycline [76]. Maximum adsorption also occurs at acidic pH values between pH = 3 and pH = 4 [66,79]. In magnetic porous chitosan-g-poly(2-acrylamide-2-methylpropanesulfonic acid) (CTS-g-AMPS), the main adsorption groups were SO_3_H and SO_3_^−^. Despite the negative charge of the adsorbent and the positive charge of tetracycline, the deprotonation of SO_3_^−^ was inhibited at pH < 3; therefore, the adsorption of tetracycline was weak. The adsorption of tetracycline was due to its interaction with the functional groups (OH, N-H, S=O, N-H, and S-O) of the adsorbent [79]. The CuCoFe_2_O_4_@Chitosan adsorbent showed that electrostatic interactions occurred between the hydroxyl and the amine groups and tetracycline [66].

In an adsorption process so dependent on the ionic interaction, it should be expected that the ionic strength can play a relevant role in the adsorption process, due to the so-called screening effect. Another non-negligible effect, as a consequence of significant ionic strength and the type of ions present in solution, is the possibility of ions competing with adsorbates in the adsorption process. For example, the presence of Na^+^, K^+^, NH_4_^+^, Ca^2+^ [71], Cd^2+^ [72], Mg^2+^, and Ga^2+^ [82] concentrations in the aqueous medium leads to a decrease in tetracycline adsorption [71,72,82]. However, several studies have shown that the presence of specific types of ions may contribute to an increase in the efficiency of tetracycline removal. This is the case with metal ions, such as Cd(II) and Cu(II), which at low concentrations form a bridge between the adsorbate and the adsorbent, increasing the amount of tetracycline adsorbed [72,74].

### 4.2. Ciprofloxacin

Similar to tetracycline, the behavior of ciprofloxacin in the adsorption process is highly dependent on pH due to its pK_a_ values.

As it is known, the surface of chitosan/biochar hydrogel beads is negative, in the pH range of 3–7, while the ciprofloxacin is positive at pH < 6.09. This justifies the maximum adsorption capacity of this material at pH = 3 [92]. The same maximum adsorption pH was found for the chitosan–poly(acrylic acid) hydrogel [105].

However, for the adsorption of ciprofloxacin on the chitosan/biochar beads, with the latter previously modified with acid humic, the maximum adsorption capacity occurs at pH = 8, which is justified by the pH*pzc* being equal to 8.3 [95]. The adsorption of ciprofloxacin onto chitosan/biochar adsorbents occurs through mechanisms such as hydrogen bonds, electrostatic interactions, π-π electron donor–acceptor interactions, and hydrophobic interactions. Ciprofloxacin has a very low solubility in aqueous solutions, and consequently the adsorption tends to occur onto the adsorbent surface and pores via hydrogen-bonding. Due to presence of -OH and -COOH groups on the adsorbent, benzene rings act as π-electron donors and π-electron acceptors. Strong electron-withdrawing fluorine and *N*-heteroaromatic rings cause the benzene ring in ciprofloxacin to act as a π-electron acceptor [92,94,95].

When the MOF–chitosan composite was used as an adsorbent for ciprofloxacin, a higher adsorption capacity was observed in the pH range of 6–9 [30,99]. At a higher pH, -COOH groups are deprotonated, and ciprofloxacin acquires a negative charge. The adsorbent and adsorbate have negative charges, leading to electrostatic repulsion between them [97,98]. The adsorption of ciprofloxacin occurred due to π-π interactions, electrostatic interactions, and hydrogen-bonding.

Some adsorbates show the maximum adsorption capacity at a neutral pH that corresponds to the zwitterionic form of ciprofloxacin. At pH < 6 and pH > 8, the adsorbents have positive and negative charges, respectively, causing electrostatic repulsion between them and ciprofloxacin [99,102,103,104]. In fact, the maximum adsorption capacity of ciprofloxacin, as well tetracycline, occurs at a pH of around 7. Under highly acidic or basic conditions, the adsorption decreases due to electrostatic repulsion, partial protonation/deprotonation, or the structural instability of antibiotics.

Adsorption of ciprofloxacin can occurs due to host–guest inclusion interactions on β-cyclodextrin cavities, which is considered a physical adsorption and is due to chemical adsorption [107,126]. Chemical adsorption occurs due to electrostatic interactions, hydrophobic interactions, and π-π interactions [99,104]. The presence of functional groups such as -OH, -COOH, and C=O on adsorbents composite with graphene oxide, and -OH and -NH on the chitosan surface, improves the electrostatic interactions between the adsorbent and ciprofloxacin. Furthermore, the π-π electron donor–acceptor interactions can be facilitated due to the presence of a fluorine atom bonded to the benzene ring and the *N*-heteroaromatic ring in ciprofloxacin. These are considered π-electron acceptors and can interact with the GO benzene ring (the -OH groups on GO make the benzene ring richer in π-electrons) [102,103].

## 5. Application of Computational Methods

In contrast with experimental work, only a few computational studies have been carried out to unveil the features of the adsorption of antibiotics on chitosan. These studies have been employed mainly to help in the interpretation of specific experimental outcomes. In order to understand the adsorption mechanism of antibiotics (such as tetracycline and ciprofloxacin) at the molecular level, some authors have employed density functional theory (DFT) [124,127,128,129]. Due to the complexity of the “real” systems involved in adsorption processes of environmental interest, computational studies usually make use of adequate simple models that can give some relevant qualitative insights [130]. In order to study the adsorption of tetracycline by a MOF–chitosan composite, Zhao et al. [124] relied on a model in which chitosan is represented by a trimer, and the Zn-based framework (ZIF-8) is composed of a four-membered ring (4-MR) window in the faces and six-membered ring (6-MR) window in the body diagonal; details on the ZIF-8 model are given in the original paper [131]. In their study, Zhao et al. [124] applied a four-step computational approach: (i) the structures of tetracycline and the chitosan–ZIP-8 model were optimized by employing DFT calculations with the B3LYP functional and the 6-31G(d,p) (or the effective core potential LANL2DZ) basis set for non-metal (or zinc) atoms; (ii) such structures were used in a docking procedure with the AutoDock vina program (Source code available from: https://vina.scripps.edu/wp-content/uploads/sites/55/2020/12/autodock_vina_1_1_2.tgz, (accessed on 2 October 2025)) [132] to identify binding structures that expected to be the most favorable; (iii) these antibiotic–adsorbent complexes were then optimized at the DFT level of theory used in step (i); and (iv) further single-point DFT/B3LYP calculations with the 6-311G(2d,2p)/LANL2DZ basis sets were employed to obtain the corresponding binding energies that also take into consideration the basis set superposition error (BSSE). It was shown from these calculations that the adsorption of tetracycline by chitosan–ZIF-8 mostly occurs on the ZIF-8 [124]. The binding energy of negatively charged tetracycline with ZIF-8 was approximately nine times greater than with chitosan. Moreover, electrostatic attraction constitutes the major contribution for the adsorption of tetracycline on the ZIF-8-chitosan composite, although hydrogen-bonding and π–π interactions may occur between tetracycline and ZIF-8. These results are consistent with the pH effect study carried out in the same work [124].

Another adsorbent containing chitosan and MOFs (ZnCo-ZIF-chitosan) which was used to remove ciprofloxacin was studied [127] by DFT/B3LYP calculations, including dispersion corrections with Grimme’s D3(BJ) method. The effect of the water solvent was accounted for by the self-consistent reaction field method based on the SMD solvation model [133]. Regarding the basis sets, the calculations employed LANL2DZ for cobalt and zinc, and 6-31G(d) for the other atoms; also, the dispersion correction was obtained with Grimme’s D3(BJ) method. The calculated Gibbs free energy for the adsorption of ciprofloxacin onto ZnCo-ZIF is significantly higher than that for other antibiotics, thus indicating a certain degree of selectivity in the adsorbent. Moreover, the DFT calculations complements X-ray photoelectron spectroscopy by showing that the ciprofloxacin adsorption occurs via strong hydrogen bonds between the -OH/-NH groups of ZnCo-ZIF-chitosan and the -COOH/-OH groups of the antibiotic molecule, as well as by π–π interactions between conjugated aromatic rings in ciprofloxacin and the imidazole rings of ZnCo-ZIF [127].

In turn, Esquivel et al. [129] combined experimental and computational approaches to study the removal of tetracycline, ciprofloxacin, and amoxicillin by chitosan that incorporates charged quaternary ammonium groups. To discover and characterize low-energy structures, they performed DFT/M06-2x calculations with the def2-SVP basis set. For the N-alkylated chitosan model, Esquivel et al. [129] used six de-acetylated glucosamine units and two cationic trimethylamine chains. The analysis of the computational results essentially relies on the molecular electrostatic potentials (MEPs) and dipole momenta of all systems in various protonation states, compatible with an interval of pH between 7 and 14. The adsorption of the antibiotics on the chitosan appears to be governed by electrostatic interactions. Although dipole–dipole interactions are the most relevant at lower pH values, due to the zwitterionic character of the antibiotics, the anionic form becomes dominant as pH increases with the resulting strong ionic attraction contributing to an adsorption enhancement, which is in agreement with the higher removal efficiencies observed in experiments [129].

In order to guide the development of new chitosan-based materials for removing zwitterionic tetracycline (zTC) from water, Sousa et al. [128] used a set of computational methods, including DFT and molecular dynamics (MDs) calculations, to study the relevant adsorption process. Chitosan (CTS) and chitosan functionalized with either 1-naphthaldehyde (RCN) or 2-hydroxy-1-naphthaldehyde (RCN1) were considered as tentative materials to interact with zTC in the aqueous environment. In this recent work [128], all structures were optimized at the DFT level by employing the B3LYP functional with 6–31G(d,p) basis set, and the D3 method of Grimme [134] with the BJ-damping approach was applied for the dispersion correction. In addition, a polarizable continuum model (PCM) was used to account implicitly for the solvent effects in the electronic-structure calculations [135]. As for MD simulations, two types of initial conditions were considered: in the first, the trajectory departs from the geometry of the zTC–adsorbent complex optimized at the DFT level of theory, while in the second, the simulation starts with the antibiotic and the adsorbent separated by about 15 Å from each other. Sousa et al. concluded that the stability of the complexes involving zTC and the chitosan materials follows the order of zTC–RCN1  >  zTC–RCN  >  zTC–CTS, and the adsorption mechanism is essentially controlled by van der Walls and π-π interactions [135]. Moreover, the computational achievements were confirmed by studying experimentally the solutions of zTC with the synthesized RCN and RCN1 polymers [135].

## 6. Conclusions and Future Perspective

The excessive and inappropriate use of antibiotics in humans, animals, and agriculture has turned these drugs into emerging pollutants. Among several available methodologies to remove antibiotics from the environment, the use of adsorbent materials has proven to be one of the most effective in reaching such goals in a sustainable way and at a low cost. Polysaccharides have been explored as adsorbents that can fulfill these requirements. In particular, chitosan is a promising adsorbent, since it is an animal-derived polysaccharide that can be recycled from crustacean waste and is very abundant, non-toxic, inexpensive, and biodegradable. As drawbacks, chitosan has poor mechanical properties (e.g., rigidity and brittleness), presents low solubility in many solvents, and is unstable in acidic medium. To address these issues, chemical modification based on processes such as cross-linking and grafting have been proposed.

In this work, we review chemically modified chitosan-based materials, including composites, that have shown to significantly improve the adsorption capacity compared to pure chitosan, particularly for removing tetracycline, ciprofloxacin, cephalexin, and azithromycin from water. Despite the great number of new chitosan-based materials that have been developed for the removal of antibiotics, very few studies can be found in the literature for cephalexin and azithromycin.

Chitosan-based composite materials with abundant functional groups and high surface areas exhibit the best efficiency in adsorbing antibiotics. Indeed, there is experimental evidence that adsorption efficiency is highly dependent on physicochemical conditions, such as pH and ionic strength, which play a significant role in controlling the interactions between the adsorbate and the adsorbent.

It is apparent from this review of the literature that, at present, only very few computational studies regarding the interaction of antibiotics with chitosan-based adsorbent molecules have been performed. Nonetheless, such studies demonstrate that computational approaches are important for unveiling details about the adsorption mechanism of antibiotics on different types of materials. For instance, it has been shown that the adsorption mechanism between chitosan-based materials and antibiotics is primarily governed by electrostatic interactions, π–π interactions, and hydrogen-bonding, particularly for tetracycline and ciprofloxacin. Thus, computational techniques develop a complementary role in relation to experimental work and, hence, should be increasingly applied in studies regarding the adsorption of antibiotics in aqueous media. As an example of this endeavor, we presented an overview of our recent computational results on the interactions between tetracycline and chitosan and chitosan functionalized with either 1-naphthaldehyde or 2-hydroxy-1-naphthaldehyde.

We should mention that, during the preparation of the present work, a review paper [136] appeared in the literature about the use of adsorption materials based on modified chitosan for removing several types of pharmaceutical residues from aqueous solutions. However, the overlap between both works is negligible regarding the pollutants that were treated. Moreover, in comparison to Ref. [136], our work gives much more emphasis to the application of computational methods. In the future, we expect that the development of sustainable and non-toxic chitosan-based adsorbents combining in vitro and in silico techniques could lead to significant advancements in environmental remediation and water treatment.

## Figures and Tables

**Figure 1 ijms-26-11070-f001:**
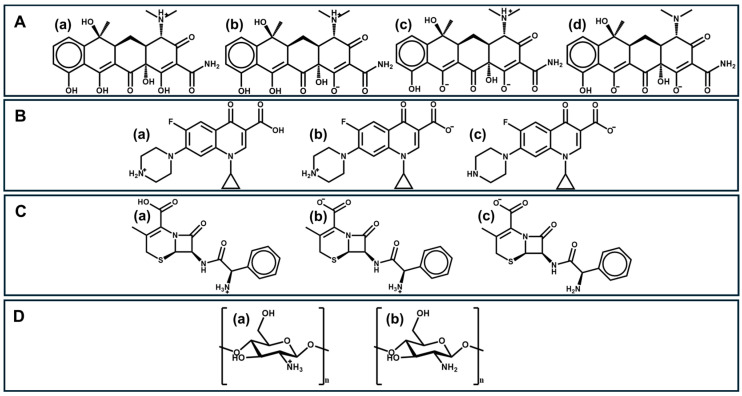
Chemical structures of (**A**): (**a**) cationic tetracycline, (**b**) zwitterionic tetracycline, (**c**) anionic tetracycline, and (**d**) dianionic tetracycline; (**B**): (**a**) cationic ciprofloxacin, (**b**) zwitterionic ciprofloxacin, and (**c**) anionic ciprofloxacin; (**C**): (**a**) cationic cephalexin, (**b**) zwitterionic cephalexin, and (**c**) anionic cephalexin; and (**D**): (**a**) non-protonated chitosan and (**b**) protonated chitosan.

**Figure 2 ijms-26-11070-f002:**
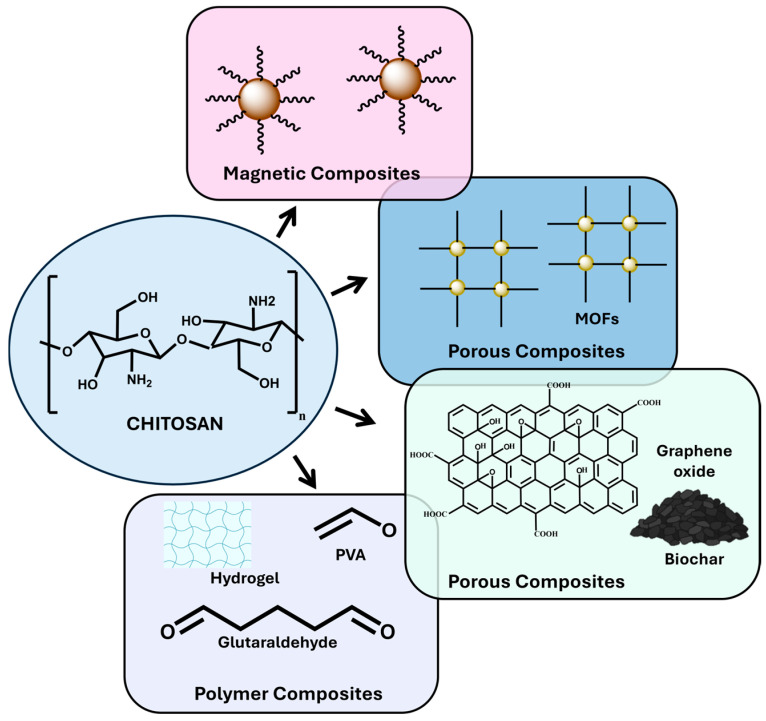
Diagram of possible materials for the formation of different types of chitosan-based composites.

## Data Availability

No new data were created or analyzed in this study. Data sharing is not applicable to this article.

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
