# Peer review of "Int. J. Mol. Sci.2025, 26(22), 11070;https://doi.org/10.3390/ijms262211070"

_ijms, 2025, doi:10.3390/ijms262211070_

Round 1

Reviewer 1 Report

Comments and Suggestions for Authors

This is a well-written literature review on antibiotic absorption on chitose-based materials.

Beyond some formatting revisions, I would suggest improving the way the data are presented, as it can sometimes be lengthy to read. Consider including some tables or graphs to summarize the findings.

The final disposal of these materials also needs to be discussed, as one drawback of adsorption is that only the liquid phase is transferred to the solid. Is this possible to implement on a large scale?

Review the entire document to correct typographical errors and carefully review the chemical formulas.

Line 52: unify the units to better appreciate the differences, for example putting everything in ng/L

Line 88, 92, etc: remove extra space

Line 142: remove italics

Line 309, 455: check subscripts

Reviewer 2 Report

Comments and Suggestions for Authors

This manuscript systematically explores chitosan-based adsorbents for removing four ubiquitous antibiotics, integrating experimental data and computational analyses to offer meaningful insights into adsorption mechanisms. However, minor revision is needed.

  1. Line 44, (Table 1), delete.
  2. The resolution of Figure 1 is low, and the letters in the structural diagram are hard to read.
  3. Line 228-231, and Line 232-235, the contents of these two paragraphs are repetitive.
  4. There's no need to list subheadings for Section 3, if there is only 3.1. Please confirm.
  5. Section 4, how stable are tetracycline and ciprofloxacin in terms of pH? Will their structures be disrupted under strong acids or strong bases, thereby losing their significance in adsorption and removal?
  6. Table 2 and Table 3 are not presented in the main text, please confirm.
  7. It is strongly recommended that Section 3 be divided into several parts. It would be better to categorize the materials based on chitosan. Currently, it is more like a collection of literature data. The third part is the section that the author needs to carefully revise.

Reviewer 3 Report

Comments and Suggestions for Authors

The manuscript by J. F. M. Sousa et al is a review on recent advances in wastewater treatment by chitosan derivatives that addresses an important and modern challenge: contamination by antibiotics. The authors collect a good library of primary research sources and put them in the context of the topic. They consider different ways of modification of chitosan macromolecules and investigate how such modifications may affect efficiency of adsorption of several groups of antibiotic contaminants including fine atomic level addressed by in silico methods. The review is stimulating and may attract the audience of International Journal of Molecular Sciences.

I found several issues at "cosmetic" level that must be addressed.

Line 109. "This review focuses on the application of polysaccharides for adsorptive 109
removal of antibiotics such as tetracycline, cephalexin, ciprofloxacin, and azithromycin 110
from wastewater." Should be more specific: "chitosan and its derivatives" instead of "polysaccharides".

Line 140. "Tetracycline, is characterized for presenting different pKa 140
=3.3, 7.7 and 9.7, showing the optimum antimicrobial activity in the pH range 5.5-6.0[49], 141
which can be justified by its zwi􀄴erionic structure (pKa =3.3, 7.7 and 9.7)." It is an awkward sentence, consider to re-phrase it.

Lines 228 - 236. Repetition. Remove one of the duplicates.

Line 281. "...have prepared an electrospun nanofibers" Article "an" should not be used here.

Line 440 and several more instances on this page:  "The maximum adsorption for tetracycline was observed in neutral solutions, between pH 6-7 ..." t should be said rather "on the pH range of ... or "between pH 6 and pH 7"

In general, I support publication of the manuscript upon minor corrections. 

Round 2

Reviewer 1 Report

Comments and Suggestions for Authors The suggestions have been taken into account, and I believe that in this way the manuscript can be published.